# Locating Hidden Elements in Walls of Cultural Heritage Buildings by Using Infrared Thermography

**Hrvoje Glavaš** [1] , **Marijana Hadzima-Nyarko** [2,*], **Ivana Haničar Buljan** [3] **and Tomislav Barić** [1]

[1]  Faculty of Electrical Engineering, Computer Science and Information Technology Osijek,
    J. J. Strossmayer University of Osijek, Kneza Trpimira 2B, 31 000 Osijek, Croatia;
    hrvoje.glavas@ferit.hr (H.G.); tomislav.baric@ferit.hr (T.B.)
[2]  Faculty of Civil Engineering and Architecture Osijek, J. J. Strossmayer University of Osijek,
    Vladimira Preloga 3, 31 000 Osijek, Croatia
[3]  Institute of Art History, Ulica grada Vukovara 68, 10 000 Zagreb, Croatia; ihanicar@ipu.hr
*  Correspondence: mhadzima@gfos.hr

**Abstract:** The structure of Tvrđa and its buildings date back to the Middle Ages. Tvrđa represents the Old Town of the city of Osijek and the best-preserved and largest ensemble of Baroque buildings in Croatia. After the withdrawal of the Ottomans in 1687, during the 18th century, the Austro-Hungarian administration systematically formed a new fortification system, regulated streets and squares and built a large number of military objects. Tvrđa took its present form in the 19th century and has kept it since then. Investigating the historical development of individual buildings, in addition to archival sources and existing architectural documentation, the obvious source of information are the buildings themselves. The aim of this paper is to explore the possibilities of using infrared thermography to find structural elements and hidden openings in historic buildings in Osijek's Tvrđa. This paper describes the exploration of the 18th century openings on the facades of the former Kostić houses. The facades were bricked into the walls in the 19th century because houses were reused and their purposes changed from commercial to residential. Infrared thermography is often a starting, nondestructive testing method (NDT) for building analyses. This paper presents thermographic analyses of two buildings. The analyses were carried out in December 2017 and January 2018. Using a steady-state thermographic analysis of a building envelope as the first step, the audit was continued with step heating (SH) of an interest point where changes in a thermal pattern were expected due to additional bricking. Heat flux was generated by the usage of a heat gun for paint removal.

**Keywords:** cultural heritage buildings; nondestructive testing method; passive thermography; active thermography

## 1. Introduction

Croatia, as a part of the Mediterranean zone of the Alpine-Himalayan seismic belt, is located in an area of high seismicity, as confirmed by earlier catastrophic earthquakes in Zagreb (1880) and Dubrovnik (1667). Old buildings, built from stone and masonry blocks, do not follow any provisions and are not in accordance with earthquake-resistant design. Therefore, it is necessary to evaluate the level of seismic risk to these old buildings. Vulnerability assessments of the cultural heritage located in seismic areas has been actively researched [1,2]. In the papers by [1,2], the authors proposed a nondestructive and relatively fast but accurate seismic vulnerability assessment of heritage buildings in Tvrđa, the old core of Osijek city. However, performing a detailed analytical design can be of great significance in order to validate this fast seismic vulnerability method. The factor that complicates seismic analysis is that heritage buildings are highly anisotropic and have complex

geometries and heavy masonry mass. The analytical seismic vulnerability method implements and takes into account more detailed analyses, such as geometrical, material and other uncertainties. It is of extreme importance to consider each detail of the structure and the difference between the structural and non-structural elements in order to generate a more realistic and accurate numerical model, which will represent a real structure. A number of parameters affect the result, such as hidden openings, which were closed in the past. This causes a large scattering effect of the results, as well as stiffness degradation.

The aim of this research is to explore the possibilities of using passive and active infrared thermography in detecting structural elements and hidden openings in historical buildings in the old city Tvrđa located in Osijek [3]. As the fourth largest city in Croatia, Osijek represents the administrative, economic, and cultural center of the Osijek-Baranja County. The old city core of Osijek, Tvrđa, is an eighteenth-century complex of cobbled streets, grand buildings and open squares, and is the most conserved set of Baroque buildings anywhere in Croatia. The two analyzed buildings are located at the addresses 23 Kuhačeva and 2 Markovićeva Street in Tvrđa. More precisely, the facades that were analyzed for the 18th century openings are oriented towards Kuhačeva Street, which is the main and most representative street in Tvrđa from the earliest days. The structure of Osijek's Tvrđa and its buildings date back to the Middle Ages. After the withdrawal of the Ottomans in 1687, the Austro-Hungarian administration systematically formed a new fortification system, regulated the streets and squares in the 18th century and built a large number of military objects. The current form of Tvrđa has not changed since the 19th century.

Due to the long period of construction of Tvrđa, it can be assumed that a large number of buildings have architectural elements from different historical periods that are often hidden or invisible today. Apart from archival research, an indispensable source of data is the buildings themselves. Apart from interpreting the present state, different types of research can be carried, the most important of which are conservative and restorative techniques for historical development. These methods can be divided into invasive, which include restoration probes, and non-invasive methods. Previous research on historical buildings has, in most cases, been destructive. By removing the original face and plaster, the wall structure would be opened in order to determine the existence of architectural elements from earlier periods. However, contemporary research on historical building development tends to use non-invasive techniques.

## 2. Methodology

Historical buildings, under a conventional thermographic survey, must have a continuous thermal flow through the outer envelope. Therefore, thermographic analysis was carried out during winter in December 2017 and January 2018. It is expected that on the structural elements of different wall thicknesses and structures, slightly different thermal patterns will be noticeable. The initial idea is to conduct a steady-state classical thermographic investigation followed by step heating of an individual location of specific interest, i.e., where hidden elements are expected to exist.

Infrared thermography presents a mature, nondestructive investigation technique that is widely used. That being said, in addition to the cost of the required camera, the main factors that led to developing a new approach to analysis are presented in this article. Thermograms were recorded with a FLIR E6 camera with a resolution of 160 × 120 pixels, field of view 45° × 34°, temperature ranges from −20 °C to 250 °C, resolution difference of <0.06 °C, refresh rate of 9 Hz and accuracy ±2% or 2 °C. Although the resolution plays a big role in choosing a camera, it is not crucial, because the spatial resolution depends on a Field of View (FOV) and Instantaneous Field of View (IFOV). FOV depends on a lens embedded in the camera. If one is planning to record power lines, it is necessary to take a camera with narrow FOV, while a wide FOV is needed if a whole building is to be analyzed. IFOV depends on the resolution because it provides information about the spatial angle of each sensor element. In order for the measurement to be accurate, the object image on the sensor must occupy a surface that is at least equal to the size or greater than one pixel of the sensing network. In the case of

our camera, a pixel would, at a distance of 1 m, cover an area of width 4.91 mm and height 4.95 mm. Thus, in order to measure accurately, the object of interest at a distance of 1 m should be greater than 9.82 × 9.9 mm, which is fulfilled in this paper.

In order to fulfill the required temperature difference between the internal and external temperatures that result in a steady state thermal flux needed for the basic thermographic analysis, the measurements were conducted during the winter period, when external expected temperatures result in a temperature difference more than 10 °C. On the places of the facade where hidden elements were expected, step heating was applied in order to obtain different thermal patterns, which indicated a difference in the wall structure, for example, air gaps. Heat flux was generated with Einhell Hot Air Gun of 2000 W adjusted to 550 °C with an air flow of 500 liters per minute.

## 3. Nondestructive Testing Methods of Buildings

### 3.1. NDT Methods

Nondestructive testing, especially of historical buildings, should address basic issues such as the role of a building, constructing material characteristics as well as characteristics of the structure that is modified by upgrades and changes of the building over time. In the absence of detailed documentation, information on the basic elements can be obtained using NDT. For the inspection of a building structure, NDT is very important for delamination, rebar location and corrosion detection [4]. NDT is also the first step in the analysis of cultural heritage based on planned further research strategies. The basic classification of the NDT test method over the past two decades has undergone changes as infrared thermography has become more available and gradually has become a standalone discipline.

The general division of the NDT method shown in Figure 1 can be further expanded, depending on specific applications. Basic nondestructive testing (NDT) methods and techniques, useful in assessing concrete structure durability, were applied for testing concrete structures in the work of [6]. One of the early Laboratory Testing Concrete Elements is available in [7]. The selection of a particular NDT method depends heavily on the object to be tested. In the article of [4], the appropriate methods for each test were provided, as presented in Table 1.

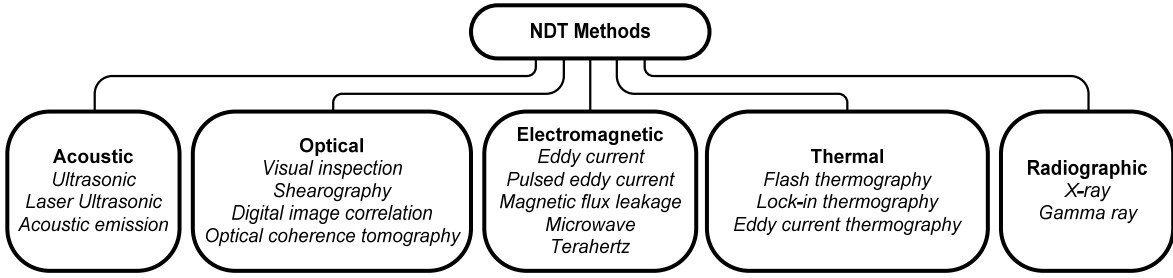

**Figure 1.** Categories of nondestructive testing (NDT) and evaluation of techniques [5].

It is common to simultaneously use multiple methods. Thus, the authors in [8] used radar and ultrasound in the analysis of overlapped bridge constructions where NDT is useful, since it causes only a limited interruption to the everyday use of the facility. Infrared thermography is often used in combination with the Ground Penetrating Radar (GPR) technique. The combination of GPR and IR thermography is described in [9] for an example of a 16th century building located in Urla, Izmir. The building is an Ottoman structure that currently houses the Urla primary school. The studies published in [10] indicate that the combined need for different NDT methods gives the best results with IR thermography due to its speed and simple interpretation, which is the first step in testing.

**Table 1.** List of items and possibilities of a nondestructive testing method usage [4].

| NDT methods \ Items | Strength | Elastic modulus | Thickness | Crack depth | Crack width | Crack distribution | Crack development | Honeycombing - voids | Lamination | Bar location | Bar size | Bar corrosion |
|---|---|---|---|---|---|---|---|---|---|---|---|---|
| Rebound hammer | ● | | | | | | | | | | | |
| Penetration resistance | ● | | | | | | | | | | | |
| Pull-out | ● | | | | | | | | | | | |
| Ultrasonic | ● | ● | ● | ● | | | | ● | ● | ● | | |
| Radar | | | | | | | | ● | ● | ● | | |
| Thermography | | | | | | ● | | ● | ● | | | |
| Radiography | | | ● | | | | | ● | | ● | ● | ● |
| Acoustic emission | | | | | | | ● | | | | | |
| Magnetic or eddy current | | | | | | | | | | ● | ● | ● |
| Half-cell potential | | | | | | | | | | | | ● |
| Photography | | | | | ● | ● | | | | | | |

The GPR technique is gaining increasing recognition in the investigation of historic buildings. The GPR technique was initially developed for geological and ground engineering research. It has proven to be a very useful means to rapidly—and nondestructively—locate metal structures such as cramps, beams, dowels and bolts within the structure of historic buildings. Particular success was also recorded in the measurement of material thickness [11].

A comparison of combined NDT techniques in civil engineering applications can be found in the following paragraphs. A laboratory and real test for concrete construction work has investigated the efficiency of GPR also applied in tandem with IR thermography and Electrical Resistivity Tomography (ERT) for the characterization and monitoring of building structures in laboratory and in-situ conditions [12]. An example from the study [10] points to finding niches using IR thermography and GPR in "Sala Delle Nicchie", which is, by age, the closest to the analyzed structures in this paper.

*3.2. Infrared Thermography*

Infrared thermography is a contactless method for determining the temperature distribution on the surface of the observed object by measuring the intensity of radiation in the infrared region of the electromagnetic spectrum. Infrared thermography is a nondestructive testing method which, due to technical advancements and the lower cost of equipment, has been extensively used. Infrared thermography is a mature technique which has become more attractive in an ever more increasing number of application fields [13]. Numerous investigations have been undertaken, some of which are mentioned for the purpose of this article. The investigation of historic structures with the use of IR thermography in order to assess the physicochemical behavior of conservation treatments such as stone cleaning, stone consolidation, repair mortars, as well as to disclose any substrate features, such as tesserae on plastered mosaic surfaces, can be found in [14]. IR thermography has been successfully applied for the knowledge of wall bonding, moisture mapping and the measure of the thermal diffusivity of bricks and plaster [15]. Mortar testing from different decades shows different behavior, as can be seen in [14]. Bonding materials come from different areas that have a material surplus, for example Spain, Portugal, Romania and Germany [16], and with their application, they can result in different thermal patterns. The detection of delamination and structural cracks of water leakage surface evaporation is described in [15]. It can be said that by using an IR camera for NDT of reinforced concrete structures, the amount of time needed to inspect the structure is significantly

reduced. This is because the result of IRT, thermograms, which screen potential concrete defects in a concrete subsurface, can pinpoint defected areas and thus reduce the amount of time to inspect compared to the sounding test, since there is no need to inspect spot by spot [17]. Other methods of NDT testing based on potential and resistivity measurements [18] require more time.

Since 2014, thermography has become widely available thanks to models placed on the market by firms FLIR, Thermal Seek and Therm App. Essentially, these are the third generation of thermographic cameras that have been developed since 1995. According to ISO 20473, thermography performs a radiation analysis in three areas (NIR 0.78–3 μm, MIR 3–50 μm and FIR 50–1000 μm), but most commonly, the classification is divided into five areas, namely a near infrared area (0.7–1,4 μm), short-wave IR (1.4–3 μm), medium-wave IR (3–8 μm), long-wave IR (8–15 μm) and far infrared area (15–1000 μm). Almost all cameras for civil use work in a long-wave IR region, except for cameras used in a gas analysis, which work in a medium-wave IR area.

With respect to the end-user, thermography can be classified as qualitative and quantitative when taking into account the information the camera provides, or passive and active when taking into account the excitement type. This paper aims to show the advantages of thermography as a nondestructive research method and to encourage the use of active thermography.

The technical characteristics of infrared cameras and the importance of using infrared thermography are most pronounced in Hong Kong [19]. Hong Kong has many buildings that are more than 40 years old, so they introduced a Mandatory Building Inspection Scheme (MBIS) in 2012. The primary step in the inspection is a visual inspection. It is followed by crack mapping, deflection measurement, settlement measurement and observations for signs of water leakage and steel corrosion. On the other hand, the condition assessment deals with sample material testing, in situ temperature measurement, moisture, half-cell electrical potential, vibration and delimitation, and occasionally even continuous monitoring. Thermography is widely used because it is contact free, can be used on large areas, is quick and can be done in real time. The major problem is qualitative, i.e., when it is applied for a surface analysis, delimitation thickness cannot be assessed, surface temperatures depend on human activities and weather, solar radiation interferes with equipment, thermal radiation can be obstructed, accuracy deteriorates with distance, the viewing angle distorts the image, and it is difficult to interpret the results due noise and variation in emissivity.

### 3.3. Qualitative/Quantitative Thermography

Qualitative thermography provides basic information about the temperature distribution on the surface of analyzed structures. The actual temperature values may differ significantly from those read on the camera. Figure 2 shows a residential building with the areas for commercial usage on the ground floor, where horizontal ties and structural elements are clearly visible, as well as a part of the staircase (bottom right corner of the building) which is not a heated space.

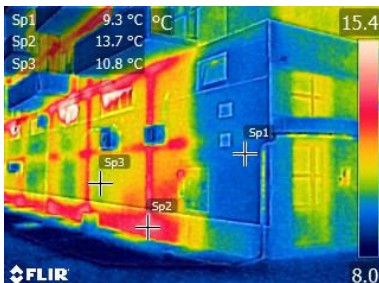

**Figure 2.** An example of highly-visible structural elements as well as a heated and an unheated space.

Based on the thermogram, it is possible to install new openings without jeopardizing the loadbearing static of the structure. Compliance with the requirements of the Technical Regulation on Rational Use of Energy and Thermal Protection in Buildings can be analyzed by infrared thermography.

Another example of an outer envelope can be seen in Figure 3, with the area where the insulation is missing colored in blue. It is evident that the wall temperature in the areas with and without insulation vary in the range of 2 °C.

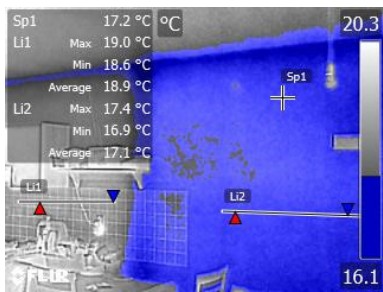

**Figure 3.** Absence of the outer envelope insulation through a homogenous thermal flow analysis.

The lack of adequate insulation on the overhead wall of the industrial facility suggests the existence of an omission that resulted in increased thermal loss (Figure 4). The cooler areas on the pavement are the parts covered with water that, by its evaporation, takes away thermal energy and ostensibly represents an area of lower temperature.

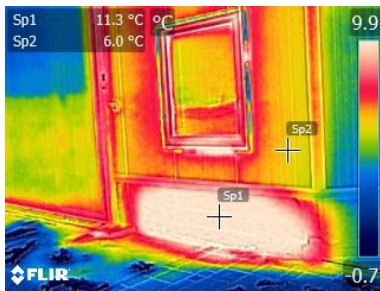

**Figure 4.** Lack of insulation of the outer envelope in front of the radiator.

The task of quantitative thermography is to provide accurate temperature values on the surface of an analyzed object. In order to complete a quantitative analysis, it is necessary to enter accurate emissivity values, apparent reflected temperatures and weather conditions when recording. The procedure of quantitative thermography requires a person with experience and knowledge gained by adopting the first degree of training. The description of the requirements for an operator of the IC camera is best described in ISO 18436-7: 2014 "Condition monitoring and diagnostics of machines—Requirements for qualification and assessment of personnel Part 7: Thermography". When performing the quantitative thermography of buildings and building openings, it should be taken into account that the emissivity changes with the angle of recording, and that humidity affects the measured values of temperature, wind, and atmospheric conditions. In the field of infrared thermography, there is no single standard. The researchers [20] used qualitative IRT tests as defined by EN 13187: 1998 (International Organization for Standardization, 1998) and RESNET Interim Guidelines for Thermographic Inspections of Buildings (Residential Energy Services Network, 2010). With respect to buildings, the most commonly used is ISO 6781-3: 2015 Performance of buildings—determination of heat, air and moisture irregularities in buildings by infrared methods—Part 3: Qualifications of equipment operators, data analysts and report writers.

### 3.4. Passive/Active Thermography

Passive thermography corresponds to the use of natural heat sources, such as solar radiation or slowly varying microclimate temperatures, whereas active thermography uses a noncontact thermal

inputs placed on the surface of the inspected body by the means of lamps, hot or cold air guns, or devices making the surface vibrate [21].

Passive thermography is a process of a thermographic analysis of structures that are in a steady state for a long period of time, i.e., in an environment where the temperature does not change. Of course, in the case of the structure in Figure 4, a temperature difference of at least 10 °C between the external and internal temperature is necessary to form a heat flow. Figure 5 shows the thermogram of the PVC opening on the outer envelope. The temperature of the structure/building is 22.9 °C; the temperature of the part of the ceiling under the infiltration of the outer air is 18.7 °C, and the reflected temperature from the glass is 19.9 °C because the glass is impervious to IC radiation, and it is difficult to measure the temperature on it correctly. The thermal reflection is best seen on the mirror image of the central heating pipe which is reflected on the surface of the glass. The line analysis on the PVC section of the carpentry shows an average temperature of 21.3 °C.

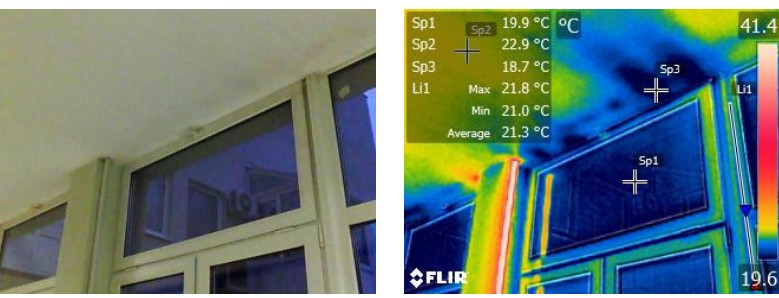

**Figure 5.** Thermogram of the openings in the outer envelope of the building.

If the outer envelope is made of various materials, the heat flow will be different with different structural elements. Figure 6 shows the outer wall of a reinforced concrete multi-storey building whose walls are made of aerated concrete units. The average temperature on the aerated concrete units is 21.1 °C, while the difference between the unit and the binder is 0.9 °C. The temperature on the reinforced concrete bearing wall is 19.4 °C, with a temperature difference of 1.2 °C.

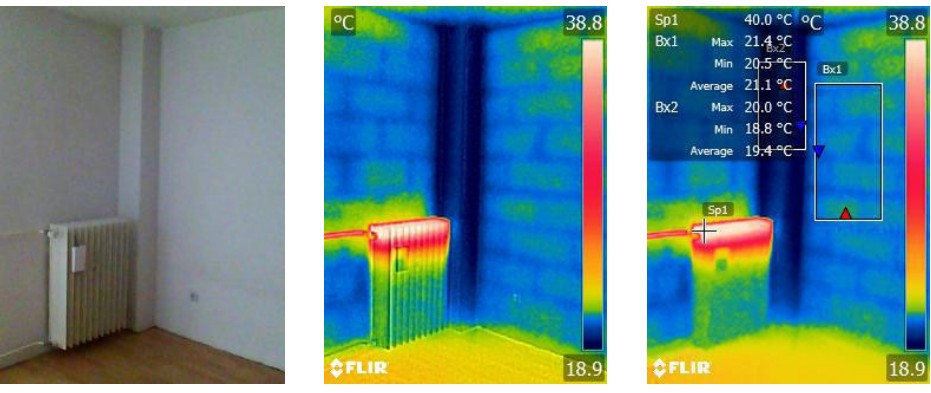

**Figure 6.** Thermal pattern of the outer wall.

A third example of passive thermography is shown in Figure 7, illustrating the section of the partition wall in the stationary state the analysis was performed in. The mean measured temperature is 29.6 °C, with a difference of 0.7 °C between the maximum and the minimum reading value being attributed to the difference in the emissivity of the wall, switches and thermostats. Reflection of the surrounding sources exists as well, but to a lesser extent. The partition wall has no heat flow, and therefore, the temperature distribution over the surface is homogeneous. In the case of an inspected body that has the same temperature as the environment, as can be seen in Figure 7, active thermography is needed.

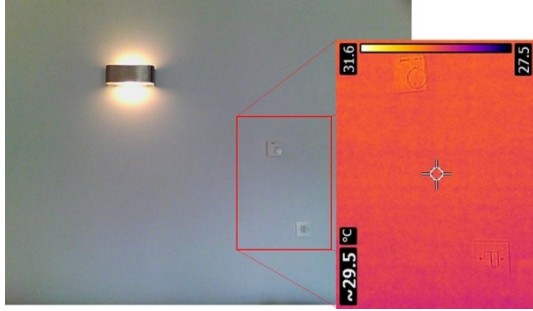

**Figure 7.** The partition wall and thermogram of the area where the hidden junction box is located.

Infrared thermal measurements, taken by applying thermal excitation to an analyzed object to achieve a temperature difference, are considered as a process of active thermography. Active thermography produces results only if the properties of an analyzed object are different from the properties of the environment where it is located. In the case of active thermography, the excitation can be ultrasonic, electric, thermal and mechanical. Thermographs are recorded in time, and on their basis and with the application of software support, information regarding material properties can be obtained. Figure 8 provides active thermography classification that can be found in [22].

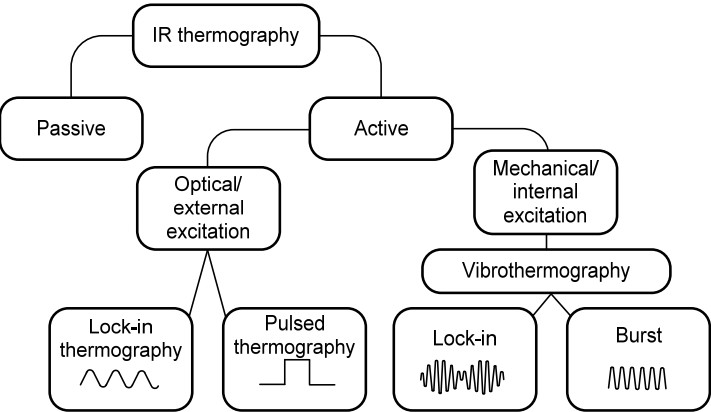

**Figure 8.** Classification of active thermography.

Active thermography nowadays refers to computer-aided thermography. Connecting the infrared camera with an excitation source via a computer and adequate software support increases the sensitivity of the measurement setup up to, on average, one hundred times (from 0.06 °C up to 10 μ°C). The disadvantage of this approach is that the measuring time (from seconds up to hours) increases with the desired thermal sensitivity. The mathematical background for active thermographic analyses in the analyzed algorithms can been seen in [23]. Using our example, the active lock in thermography, with optical external excitation, could provide information about deeper layers of facade. Pulse thermography is also applicable, but due to lower thermal conductivity, thermal impulse is longer. An example of pulse thermography can be seen in [24]. In our case, the possibility of applying active thermography is reduced because of the large analyzed area, its frequent use by pedestrians and the absence of the required equipment.

Based on the previous experience of the application of step heating results, we were encouraged to replicate the procedure on historical buildings. For example, the case of a hidden box, in which a thermostat connection is performed, is presented on the thermogram in Figure 9. The box position is not known, but it is assumed that it is near the switch. In order to determine its exact position, a hair dryer can be used to heat the wall. The thermographic camera is used to monitor the response to the heat excitation. The surface behind which the box is hidden has increased its thermal resistance and heat retention, which is longer in duration due to the difference in thermal resistance. The initial

location of the hair dryer is presented in Figure 9a while the indication of the box position, as well as the structure of the bearing switch elements, can be seen in Figure 9b. Over time, the outline of the box contour can be clearly seen in Figure 9c.

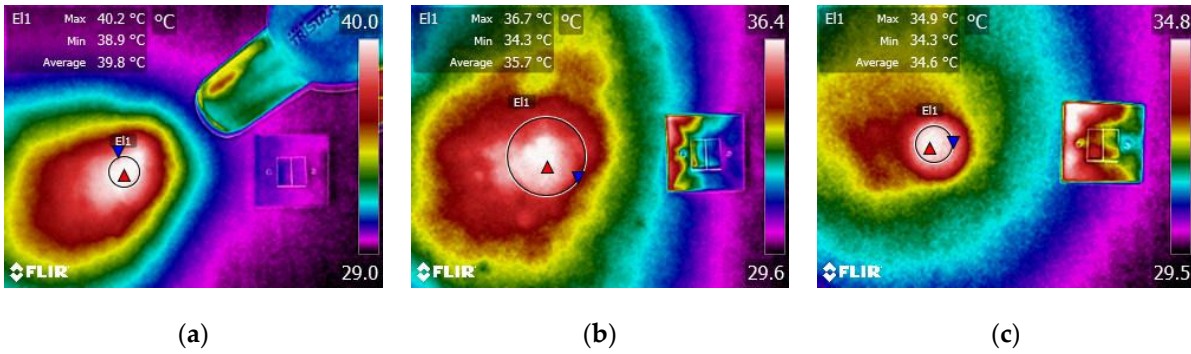

(**a**)                                (**b**)                                (**c**)

**Figure 9.** Finding a hidden distribution box by means of thermal incentive.

The time duration and the temperature values are shown in Figure 10. The measured values, marked with squares, represent the point which is located between the box and the switch, while the line, marked with diamonds, indicates the temperature in the area of the junction box.

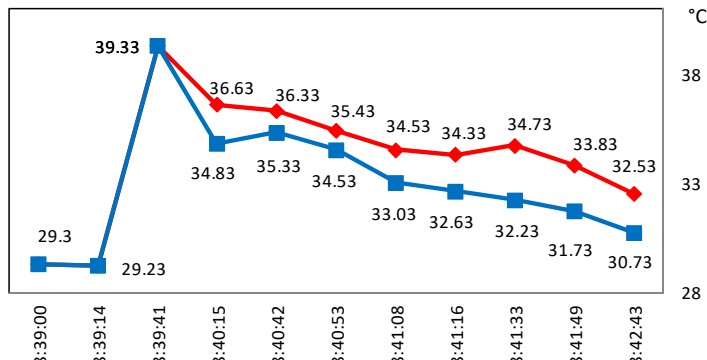

**Figure 10.** The temperature values in the junction box and its immediate area.

Figure 10 shows the precise temperature values in the analyzed spot point, but not their spatial distribution. Due to camera accuracy and slight camera movement, individual temperature measurement readings over time tend to show variations, although the actual temperature values are continuously falling. This is one of the basic problems associated with classical thermography. The special temperature distribution is presented in Figure 11, where the hidden junction box is located, which clearly illustrates the location of the connection line by which the thermostat is connected to the junction box.

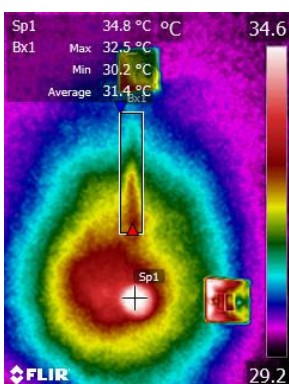

**Figure 11.** Thermogram of the hidden junction box.

In Figure 11, the position of the conductor under plaster is visible due to the material variation. It should be taken into account that, during the recording, the observed image is monochromatic and that, in most cases, it represents the sum of radiation in the IR spectrum. The advantages of using an IR thermal camera in finding a power supply cable can be seen in Figure 12, where materials of similar characteristics were used, namely plaster and gypsum.

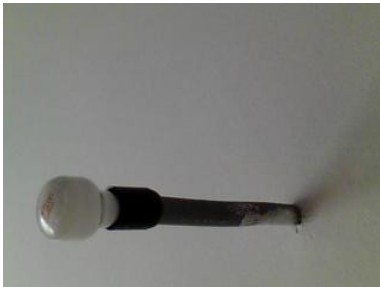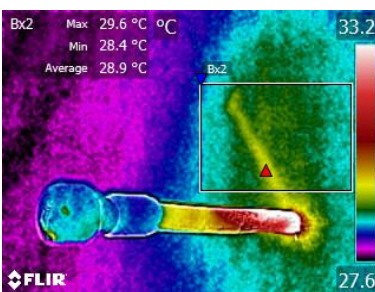

**Figure 12.** Finding a power supply cable with active thermography.

The position of the power supply cable can be often observed by applying thermal excitation. When installing a lighting fixture, the danger accidental mechanical damage of a power supply cable is, in that case, reduced to a minimum. Encouraged by the previously described experiments with thermal step excitation, the idea for research regarding finding hidden elements in historical buildings was initiated.

### 3.5. Parameters Affecting Radiometric Accuracy of IR Thermography

There are a number of parameters that ultimately define the accuracy and repeatability of a radiometric measurement. These include an object, surface, thermal environment, atmosphere, and instrument. Consideration should also be given to the measurement method or instrument that is establishing the deviation, as it may be the source of an error rather than the infrared instrument.

Surface temperature considerations depend upon a type of a thermal generator. There are three fundamental types of internal thermal generators, namely (i) a constant temperature generator example electrical fault, (ii) a constant power generator example building, and (iii) a finite energy source, for example, a motor that has been switched off. Heat transfer from these fundamental types of thermal generators can be complicated over a period by thermal capacitance(s) of the material between the energy source and the surface. An overview of the material characteristics and the typical emissivity necessary for the implementation of a thermographic analysis is given in Table 2 [14].

**Table 2.** Thermo-physical-optical properties of various materials [14].

| Material | Density $(kg\,m^{-3})$ | Specific Heat $(J\,kg^{-1}\,K^{-1})$ | Thermal Conductivity $(W\,m^{-1}\,K^{-1})$ | Thermal Diffusivity $(\times 10^{-9}\,m^2\,s^{-1})$ | Thermal Effusivity $(Ws^{0.5}\,m^2\,K^{-1})$ | Emissivity $(\lambda = 8–12\,\mu m)$ |
|---|---|---|---|---|---|---|
| Limestone | 2600 | 920 | 2.1 | 877.92 | 2241.25 | 0.93 |
| Plaster | 1440 | 800 | 0.5 | 434.03 | 758.95 | 0.91 |
| White marble | 2695 | 870 | 3.14 | 1339.22 | 2713.33 | 0.95 |
| Grey marble | 2650 | 870 | 6.7 | 2906.09 | 3930.25 | 0.95 |
| Cement marble | 3100 | 840 | 0.85 | 326.42 | 1487.75 | 0.86 |
| Concrete | 2400 | 1008 | 1.65 | 682.04 | 1997.92 | 0.94 |
| Red brick | 2025 | 800 | 0.6 | 370.37 | 985.90 | 0.90 |
| Air | 1.16 | 1007 | 0.026 | 22257.98 | 5.51 | - |
| Water | 1000 | 4193 | 0.586 | 139.76 | 1567.51 | 0.96 |

The characteristic behavior of the masonry wall described in [25] provides an initial estimate of the optimal time during the year to begin testing, taking into account the necessary difference in temperature to achieve a homogeneous thermal flow. For analytical purposes, the test was carried out

with a temperature difference (ΔT) over the building facade of at least 10 to 15 °C to allow measurable heat exchange through the element [20].

As shown in Figure 13, the wall temperatures varied by up to 55 °C or more on the surface, decreasing below freezing on cold winter nights. This wide discrepancy should be compared with a 12 °C seasonal variation in the wall center and 2 °C in the interior the latter, which probably receives some heating in winter. Daily variations of the surface temperature will be greater in summer, e.g., 42 °C, compared with winter, e. g. 25 °C. The dew point for interstitial condensation is located near the outer surface. Consequently, this can result in freezing in the outer layers, since a great deal of heat is required to raise the temperature of a heavy building with thick walls by even 1 °C, i.e., it takes time for the heat to flow through the brick and mortar. There is a tendency for the outer face to be heated up prior to a response the core can provide and long before the inner face is affected. This time lag would affect the theoretical temperature gradient.

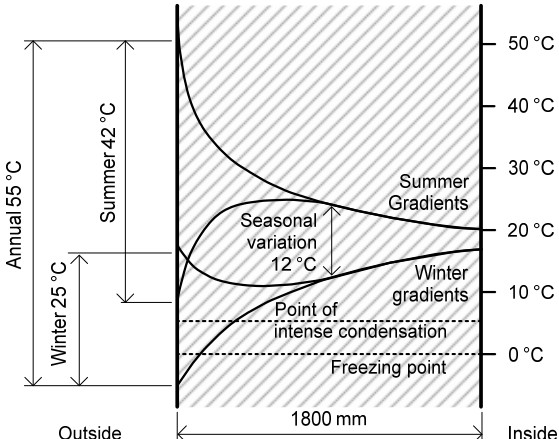

**Figure 13.** Theoretical temperature changes in a thick wall [25].

## 4. Study Area

*Analyzed Elements of Kostić House*

On the basis of the purchase contract with the Stekić family, between 1733 and 1747, trader Mihovil Kostić became the owner of the house in 23 Kuhačeva Street and the house on the corner of Kuhačeva and Marković Street (2 Markovićeva Street) (Figure 14a) [26]. Both buildings are two-storey houses with gabled roofs, masonry brick and stucco. In the architectural layout from 1774, when the military government of Tvrđa bought the Kostić houses, spatial organization and facade partition are evident. On the ground floor, there were shopping spaces with shop windows/openings; the door and the window were joined by a unique segmental overhang. There were two shop openings according to the drawing, east and west of the centrally located entrance on the ground floor. The position of these openings in 23 Kuhačeva Street was confirmed by the restauration probes (Figure 14b).

After the military government bought Kostić houses in 1774, several restructurings in the late 18th and early 19th century were carried out. On that occasion, shop openings were closed because of the reuse of the store spaces, i.e., into a dwelling on the ground floor of 23 Kuhačeva Street and into the post office in 2 Markovićeva Street. Based on drawings from 1774 (Figure 15) which depict shop openings on the facades of brick houses oriented to Kuhačeva Street, and the discovery of the same openings by a construction probe on the ground floor of the house in 23 Kuhačeva Street, the aim of the article was to investigate the existence of shop opening from the 18th century on the house in 2 Markovićeva Street through non-invasive methods.

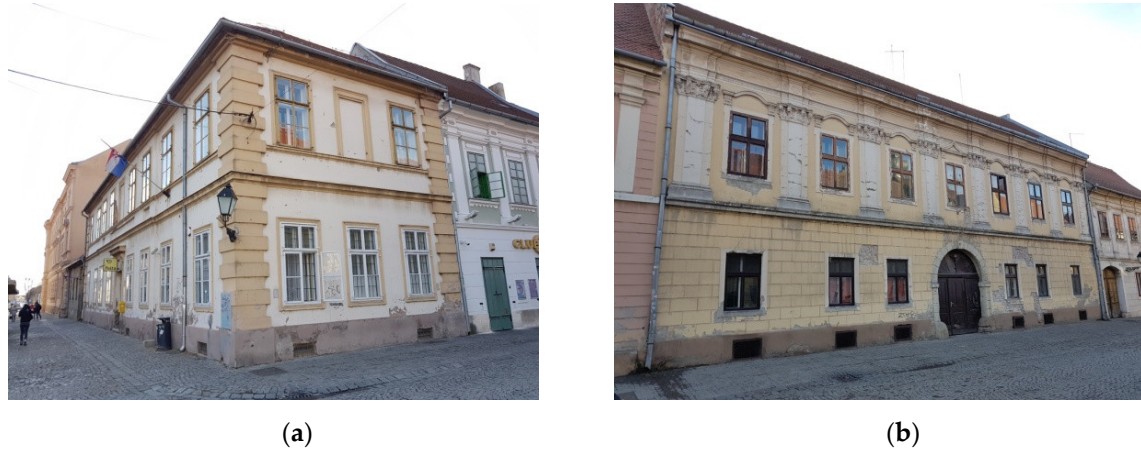

(**a**)　　　　　　　　　　　　　　　(**b**)

**Figure 14.** Kostić house (**a**) at the corner of Kuhačeva and Marković street (2 Markovićeva Street) and (**b**) in 23 Kuhačeva Street.

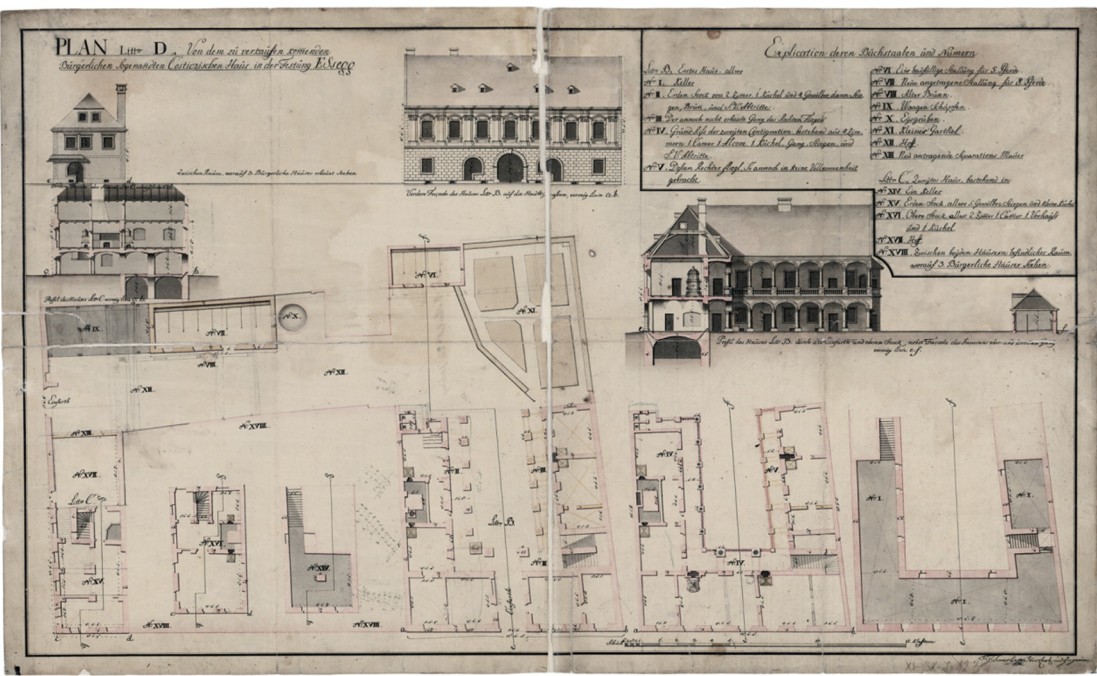

**Figure 15.** Architectural drawings of Kostić houses in 23 Kuhačeva Street from the 1774 [27].

## 5. Results

*Thermographic Analysis of Kostić Houses*

A thermographic diagnosis of historical buildings requires a multidisciplinary approach. For a high quality thermographic analysis of the buildings, it was necessary to carry out a survey of the building of interest. The complexity of the thermographic analysis process is highlighted in the provided investigation of the older buildings [17]. So far, the presence of concrete deterioration, water seepage, cover delamination and significant cracks was investigated in the work of [9]. In addition, the existence of delamination and detachments of the murals were performed as well. The thermographic analysis of Kostić house on the corner of Marković and Kuhačeva Street was carried out on 19th December 2017 between 2:15 PM and 2:45 PM. The outer temperature ranged from 1.7 °C to 1.5 °C with a relative humidity ranging from 63% to 75%. Figure 16 shows a thermogram of the outer envelope. The analysis was performed with the setting of the emissivity 0.9 according to the preposition put forward in [28]. When the exact temperature value of the analyzed thermal radiation

has to be determined, the important factor is emissivity. But, in this case, that was the aim of the analysis. For the purpose of finding discontinuity in building elements, the temperature difference is important. Although some basic guidelines for emissivity values are given in Table 2, more detailed data can be found in Table 3. Table 3 shows typical emission values of building materials.

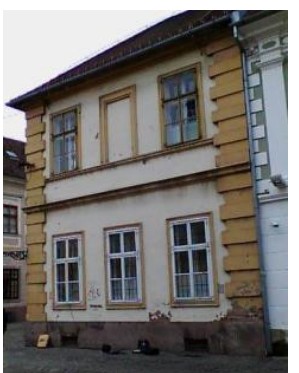 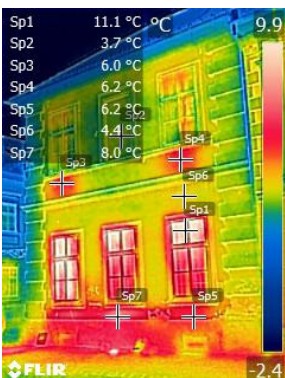

**Figure 16.** Thermogram of the whole building.

**Table 3.** Emission values for typical building materials [29].

| **Brick: Common** | **0.81–0.86** |
| --- | --- |
| Brick: common, red | 0.93 |
| Concrete | 0.92 |
| Concrete: dry | 0.95 |
| Concrete: rough | 0.92–0.97 |
| Mortar | 0.87 |
| Mortar: dry | 0.94 |
| Plaster | 0.86–0.90 |
| Plaster: rough coat | 0.91 |

The wall thickness of the analyzed building is 74 cm, and the thickness of the parapet below the window, where the heating elements of the city's central heating are located, is 36 cm. From the thermogram in Figure 16, it can be seen that the building is heated at the project temperature. The thermogram also shows the position of the radiators on parapet walls under the window. In Figure 16, the thermal image shows that the interior windows on the ground floor are open, since the space is used for the post office, and a greater amount of fresh air is required.

The outer envelope (i.e., the facade) has not been renewed since the 1980s. An additional analysis on the parts of the envelope with visible damage was carried out, but the conclusion that can be provided is that the shape of the damage does not strictly correlate with the thermal pattern (Figure 17).

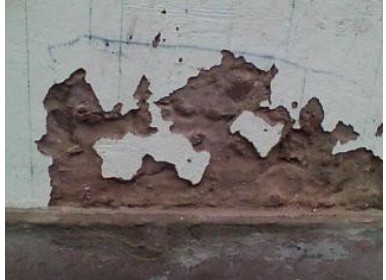 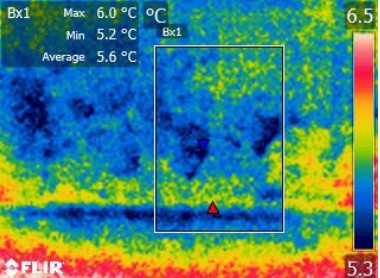

**Figure 17.** Analysis of facade damage.

A weak dot heat bridge on the first floor was noticed (Figure 18). This was probably a result of the metallic anchorage (structural element) covered with plaster during the last facade maintenance.

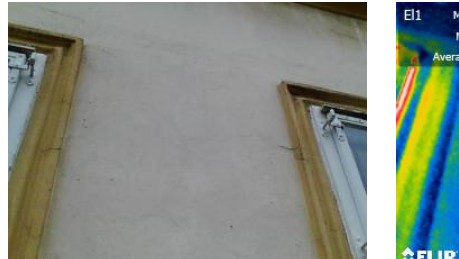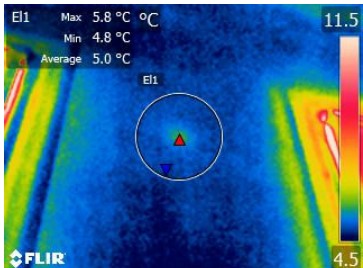

**Figure 18.** Discovered weak point heat bridge.

In order to determine the existence of discontinuity in the outer envelope, heating was applied, as is shown in Figure 19.

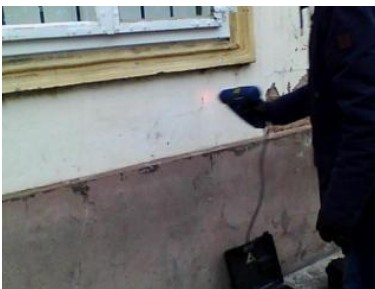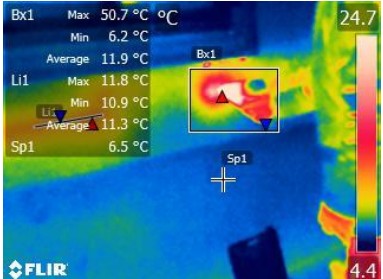

**Figure 19.** Initial application moment of thermal excitation.

There was not any discontinuity by the analysis of the thermal pattern change. In Figure 20, a linear trace of the heating body, which is continuous and uninterrupted, can be observed.

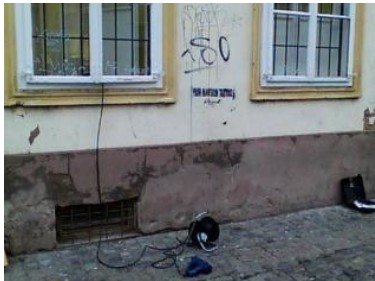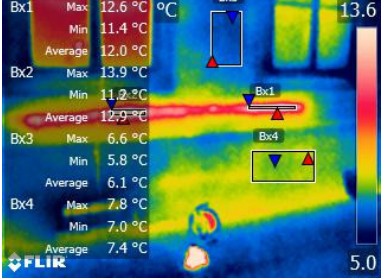

**Figure 20.** Homogeneous thermal pattern on excitation along the wall.

The detail below the left window in the lower left corner of the thermogram reveals that spot facade damage, due to a difference in emissivity, shows a higher apparent temperature (Figure 21).

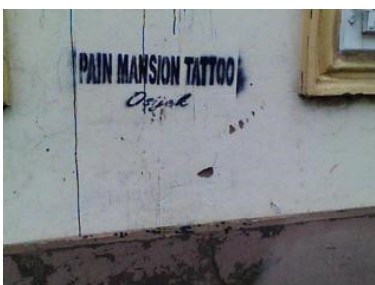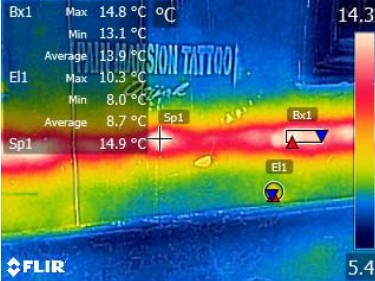

**Figure 21.** Influence of damage to the thermal pattern.

Increasing the area under the thermal excitation did not affect the result. The thermal pattern caused by the excitement was homogenous and uniformly disappeared over time (Figure 22).

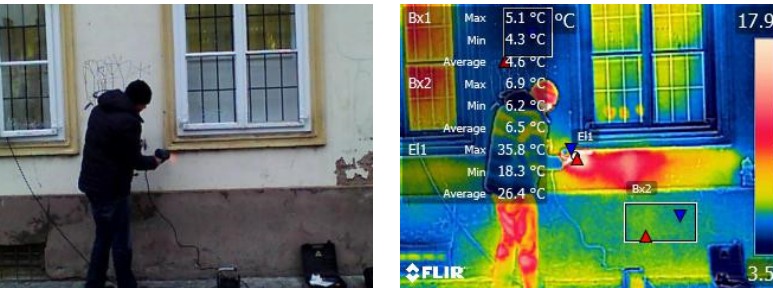

**Figure 22.** Increasing the area of applied thermal excitation.

As no discontinuity in structural elements of a building was discovered, the conclusion was that there was no change of the analyzed area, or the method itself is not applicable because of the wall mass and thermal capacity. In the case of heating the lightning protection grounding line, the heating capacity of the wall and the surface character of the applied method were observed (Figure 23).

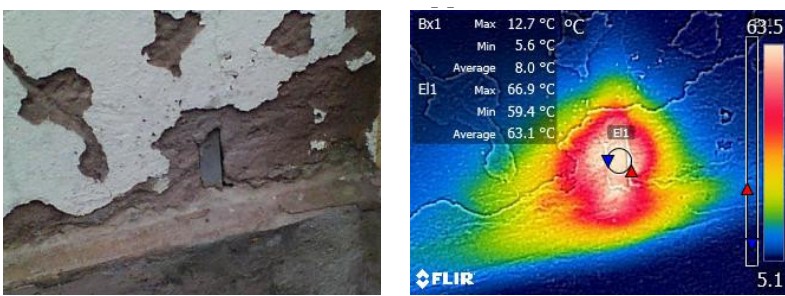

**Figure 23.** Thermal excitation of lightning protection installation.

The measurement results were obtained based on nine thermograms and the analysis of 39 measuring areas, yielding the maximum, minimum and average values of the various elements of the outer envelope and the heated area. The results, presented in Table 4, represent the average values of maximal, minimal and average values for all multiple measurements. The parapet wall is separated as the thinest part of the envelope, which contains the heating elements connected to the town central heating system.

**Table 4.** Characteristic values of the measured temperature.

| Heading | Max. Value °C | Min. Value °C | Average °C |
|---------|---------------|---------------|------------|
| Wall | 5.5 | 4.9 | 5.2 |
| Wall parapet | 6.7 | 6.1 | 6.5 |
| Window | 8.5 | 6.7 | 7.7 |
| Heated area | 14.4 | 11.8 | 13.3 |

It was decided to proceed with the analysis on the second building where the probes were made and there was existing documentation of the space reusage. Testing at Kostić house in 23 Kuhačeva Street was conducted on 7th February 2018 between 12:51 AM and 1:04 PM. The temperature at the time of the analysis ranged from 9.1 °C to 9.9 °C, and the humidity from 81% to 83%. The initial state is visible on the thermogram in Figure 24. The registered thermal radiation corresponds to the air temperature.

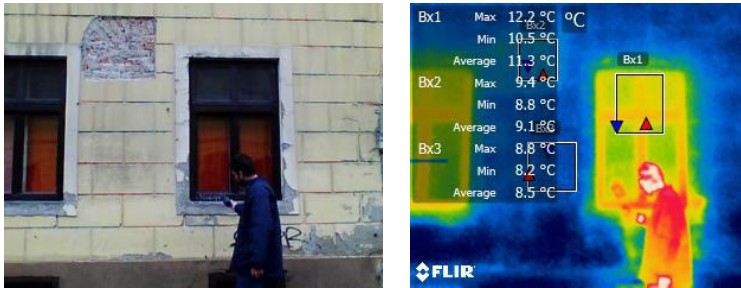

**Figure 24.** Termogram of Kostić house in 23 Kuhačeva Street.

By analyzing the thermal pattern shown in Figure 24, the difference in the probing area, as a result of the difference in emissivity, is evident. The expected discontinuity observed by probing did not appear during the heat excitation of the parapet wall (Figure 25a). As can be seen from Figure 25b, a further line heating process raised the wall temperature but did not result in a change in the thermal pattern. During the heating and during the time, the thermal pattern seemed to be inhomogeneous, albeit in a manner that corresponds to the presence of moisture, rather than non-discontinuity in the structural elements. By analyzing the smaller parts of the wall, no additional information (as shown in Figure 25c) was obtained.

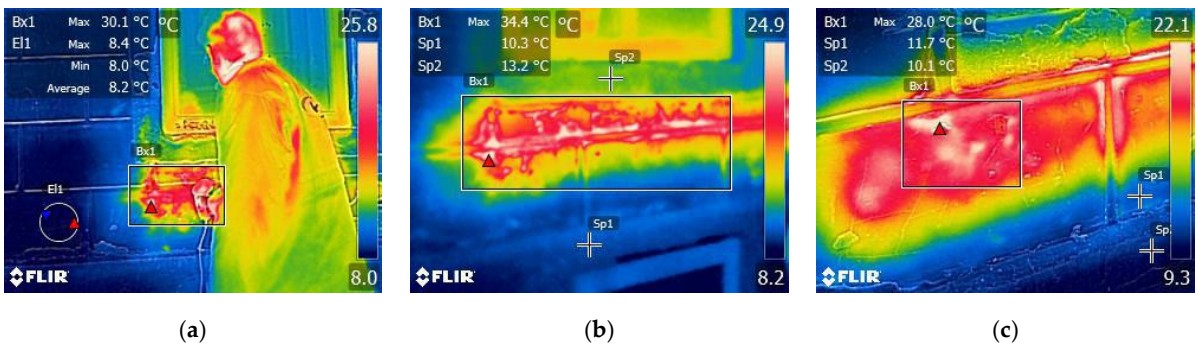

(**a**)  (**b**)  (**c**)

**Figure 25.** Changing the thermal pattern due to heating.

After looking at the bricked up openings as well as wall probing (Figure 26) and the associated thermal pattern, it is obvious that the thermographic analysis was not as effective in finding the bricked up openings in the massive homogeneous structures of the brick wall.

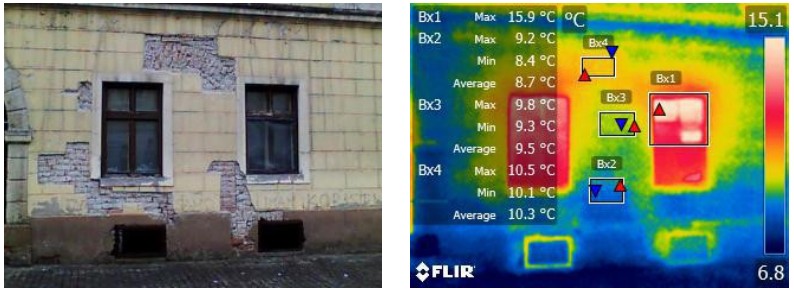

**Figure 26.** Thermogram of the probing building part of a known wall structure.

In order to confirm the conclusions, the examined place was checked from the inside of the building. Significant thermal masses and continuous heating have resulted in a constant thermal pattern, presented in Figure 27.

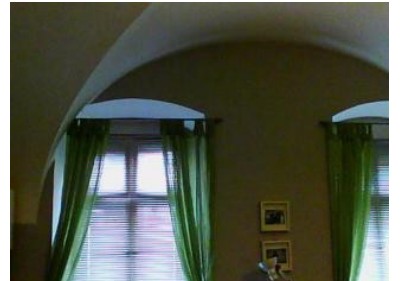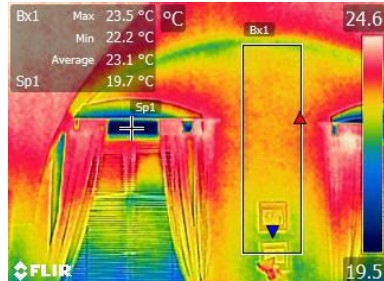

**Figure 27.** Thermal pattern on the inside of the wall.

The additional measurement results are obtained on eight thermograms and the analysis of 22 measuring areas which resulted in the maximum, minimum and average values of the various elements of the outer envelope and the heated area. The results, presented in Table 5, represent the average values of maximal, minimal and average values for all multiple measurements.

**Table 5.** Characteristic temperature values of the outer envelope.

| Heading | Max. Value °C | Min. Value °C | Average °C |
|---|---|---|---|
| Wall outside | 9.6 | 9.0 | 9.2 |
| Wall inside | 22.3 | 21.5 | 21.6 |
| Window | 12.7 | 11.5 | 12.0 |
| Heated area | 27.9 | 14.1 | 18.8 |

Unlike the first building in 23 Kuhačeva Street, there is no town central heating. Therefore, the table does not have the temperature value of the parapet wall.

Based on the provided experimental results presented in this article, conclusions are similar to those in [30], where the authors concluded that the conventional thermographic techniques are useful in superficial and subsurface testing up to several millimeters in depth and that anomalies occurring in deeper layers may be effectively located and identified by means of GPR. Dynamic thermographic methods reveal the outlines of anomalies present in an inner structure of walls, located even under 1.5 cm of plaster. However, no voids were found in this case, as in [31], and the temperature pattern shows homogeneous behavior.

## 6. Further Investigations

Due to institutional equipment availability, using GPR, further tests will be performed as in [32]. The research carried out by [32] demonstrates the suitability of GPR for identifying void spaces when running across a rough-surfaced wall facade. This is done by comparing three commonly-used antennas (1.2 GHz, 1.6 GHz and 2.3 GHz). GPR can identify features within the blocks; however, without having supplementary information from secondary sources, it is impossible to confirm the identity of the features.

## 7. Conclusions

IR thermography, as a nondestructive testing method, can easily be applied to help detect non-homogeneous outer shell elements, hidden openings and structural elements. The main characteristic of IR thermography is the detection of radiation on the surface of an object. In order to analyze the deeper layers, it is necessary to use active thermography or some other NDT method. Therefore, it can be concluded that conventional steady-state IR thermography can be useful to perform the first initial test method, which is then followed by other testing methods.

Proper interpretation of the thermogram depends on the knowledge and experience of the operator during the process of analysis. The operator is very important since, by understanding

the physical behavior of the object, he/she relates information about the state of the object and thermographic patterns. Based on this, it is obvious that for a quality thermographic analysis, a good understanding and knowledge of the historical development of the analyzed object is needed, alongside a basic knowledge of the thermal camera's operation. With the arrival of cost-effective thermocameras on the market, classical thermography is becoming far more widespread, but active quantitative thermography still requires significant financial investment in the equipment and training of operators. Currently, various active thermography methods represent the latest developments in the field of IR thermography; however, even if we neglect the financial needs for the equipment, they require more time to conduct measurements.

Due to limited financial sources, an affordable thermography camera with heat flux step excitation was used to implement this basic idea to find discontinuities inside structural walls. As preparation for the described building analysis, finding a junction box, as a typical example of this technique, was performed. Due to the different heat resistance and heat capacity of the analyzed part of the wall surface, in a relatively short time interval, it was possible to detect hidden elements in the wall. In the aforementioned example, a hair dryer was used as an exciter, but a variety of heat sources are used in everyday practice.

In the case of the heat flux applied to the analyzed buildings, classical IR thermography did not prove effective in the process of finding structural elements in the historic structures of the old city of Tvrđa. From the recorded measurements, it was found that the main reason that classical IR thermography failed was the similar amount of thermal conductivity of the materials used, as well as the wall thickness, which results in thermal haze and the equalization of thermal patterns. The test was carried out in a steady state with a constant thermal flow resulting from a temperature difference of 20 °C. No discontinuities, due to air gap, significantly different materials or elevated humidity differences, were found by heating the surface with a heat gun and analyzing the thermal conductivity over time. By heating the lightening protection installation, the thermal capacity of the wall was confirmed. It can be concluded that, as a final remark of this paper's aims, the conducted tests corresponded to the experiences presented in the analyzed literature.

**Author Contributions:** Conceptualization, H.G. and M.H.N.; Methodology, H.G.; Validation, H.G., M.H.N. and T.B.; Formal Analysis, H.G., M.H.N. and T.B.; Investigation, H.G., M.H.N., I.H.B. and T.B.; Resources, I.H.B. and H.G.; Data Curation, H.G. and M.H.N.; Writing—Original Draft Preparation, H.G.; Writing—Review & Editing, M.H.N.; Project Administration, I.H.B.

**Funding:** This research received no external funding

**Conflicts of Interest:** The authors declare no conflict of interest.

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
