# Peer review of "Locating Hidden Elements in Walls of Cultural Heritage Buildings by Using Infrared Thermography"

_buildings, doi:10.3390/buildings9020032_

Reviewer 1 Report

The article covers the topic of the location of hidden elements in walls of cultural
heritage buildings using infrared thermography. According the authors, infrared thermography  
is helpful to exploration of the 18th century buildings.
Some suggestions to enhance the paper quality are as follows:
1. The abstract should contain at least one clearly defined proposal of performed analysis.
2. The introduction is too extensive. I suggest to remove fig.1, (fig. 1 and 2 contain the similar content).
3. In the table 1 change "strengt" to "strength".
4. The paragraph in lines 65-84 should be moved to the point 2. Infrared thermography should be describe in this point.
5. In the point 2, consider to add the sub point 2.1 'Ndt methods', where the infrared thermography will be described.
6. As authors mentioned in the point 5, where it has been highlighted that the future tests will be performed using GPR method, the description in lines  117-131 should be less detailed.
7. Article could contain paragraph 'methods' where the specification and parameters of used equipment will be shown (ex. lines 156-168).
8. The point 'methods' should contain a detailed description (subpoint) of the test conditions (it should be moved from the point results). Results should contain results only.
9. The citied references are too wide. I suggest to reduce the number of citied articles up to 30.
10. If research were supported by grant etc., the founding should be contained.
I suggest that article could be considered to publication after minor revision due to necessary corrections/improvements.

Author Response

Dear reviewer, 

thank you very much for the detailed analysis of our paper. 

Your remarks were of great help to see the methodological omission.

The answers according to the suggestions are attached.

Sincerely,

Marijana Hadzima-Nyarko

Reviewer 2 Report

The document describes the exploration of 18th century openings on the facades of old Kostić houses, which in the 19th century were walled into the walls due to the reuse of houses from commercial to residential ones.  However, a very important part of the article is dedicated to a very general overview of NDT&E methodologies and a general description of thermography, and only a very limited part to the specific analysis of what is described in the title.

The expression “testing  methods  (NDTs)  that  passively  analyzes” in the abstract seems to be contradictory to what was previously written: ”infrared thermography, passive and active”.

Please clarify or modify.

Most of the problems described in the part of the article dedicated to measurements are well known problems of passive or step heating active thermography in the presence of thick walls: specific techniques have been developed to overcome these problems (i.e. thermal noise; dependence on environmental or seasonal thermal conditions), also adopted in the field of Cultural Heritage preservation. See for example [1-3]. Although it is true that 'active quantitative thermography still requires significant financial allocations for the equipment and training of operators', at least referring to such ways of solving these problems would be necessary.

Moreover, having neglected these technologies, it makes questionable some of the conclusions that seem to underestimate the potential of thermographic techniques. 

Finally, please correct the typos on lines 86, 101, 380, 416.

References

[1]Sfarra, S., Cicone, A., Yousefi, B., Ibarra-Castanedo, C., Perilli, S., & Maldague, X. (2019). Improving the detection of thermal bridges in buildings via on-site infrared thermography: the potentialities of innovative mathematical tools. Energy and Buildings, 182, 159-171.

[2]Laureti, S., Silipigni, G., Senni, L., Tomasello, R., Burrascano, P., & Ricci, M. (2018). Comparative study between linear and non-linear frequency-modulated pulse-compression thermography. Applied Optics, 57(18), D32-D39.

Author Response

Dear reviewer, 

thank you very much for the detailed analysis of our paper. 

Your remarks were of great help to see the omission in clarifying applied method. 

We have additionally explained the definition of active thermography, referred to suggested literature, explain the benefits of active thermography in increasing sensitivity, clarified our approach in conducted analysis and corrected conclusion. 

Hope that we understood and made all the changes that was suggested by you and other reviewers.

The answers for the suggestions are attached.

Sincerely,

Marijana Hadzima-Nyarko

Reviewer 3 Report

The manuscript buildings-428332 titled “ Location of hidden elements in walls of cultural 2 heritage buildings using infrared thermography” consist of 22 pages, 32 figures, 5 tables and 44 references. The structure of the manuscript is appropriate and the manuscript itself is innovative. The subject of matter is within the scope of the journal.

In the introduction all necessary information  are presented and the introduction is well organized. Presented literature is limited to the newest knowledge and is mainly from 2000 till 2018. The methodology is sufficiently explained that someone else, who has the equipment and required knowledge about the topic, could repeat the study. The conclusions are presented clearly and compared with other studies.

Although some improvements might be implemented:

-Line 30: “[1-6]” instead of “[1], [2], [3], [4], [5],[6]”

-Line 86: „Non-destructive testing of buildings of especially historical buildings” sounds strange. Maybe “Non-destructive testing especially of historical buildings” is enough.

-Line 107: Strength instead of “Strengt”

-Line 278 ( Figure 11 ): Why the temperature rises to the 34.73 from 34.33 at 8:41:33, if the heating source was switched off.

-Figure 15 and 16 could be presents as one with a) at the corner of Kuhačeva and Marković streets (Markovićeva 2) and b) in Kuhačeva 23

-Figures 21 and 27 are similar and if there is not a thermograph in the figure 27, it might be deleted.

Some figures as for example 28, 29 and 30 might be presented as one because they investigated object is the same.

-the description of the figures might be written in comparative manner and joined in longer paragraphs

Taking above into consideration there are minor issues which might be easily corrected by the authors and manuscript in my opinion should be published.

Author Response

Dear reviewer, 

thank you very much for the detailed analysis of our paper. 

Your remarks were of great help for paper improvement.

The answers to the suggestions are attached.

Sincerely,

Marijana Hadzima-Nyarko

Round  2

Reviewer 2 Report

the authors have sufficiently taken into account the indications of this reviewer